# The Interplay between PARP Inhibitors and Immunotherapy in Ovarian Cancer: The Rationale behind a New Combination Therapy

**DOI:** 10.3390/ijms23073871

**Published:** 2022-03-31

**Authors:** Brigida Anna Maiorano, Domenica Lorusso, Mauro Francesco Pio Maiorano, Davide Ciardiello, Paola Parrella, Antonio Petracca, Gennaro Cormio, Evaristo Maiello

**Affiliations:** 1Oncology Unit, Foundation Casa Sollievo Della Sofferenza IRCCS, San Giovanni Rotondo, 71013 Foggia, Italy; davideciardiello@yahoo.it (D.C.); e.maiello@operapadrepio.it (E.M.); 2Department of Translational Medicine and Surgery, Catholic University of the Sacred Heart, 00168 Rome, Italy; 3Gynecologic Oncology Unit, Catholic University of the Sacred Heart, Scientific Directorate, Fondazione Policlinico “A. Gemelli” IRCCS, 00168 Rome, Italy; domenica.lorusso@policlinicogemelli.it; 4Division of Obstetrics and Gynecology, Biomedical and Human Oncological Science, University of Bari “Aldo Moro”, 70121 Bari, Italy; m.maiorano23@studenti.uniba.it (M.F.P.M.); gennaro.cormio@uniba.it (G.C.); 5Oncology Unit, Department of Precision Medicine, Università degli Studi della Campania “Luigi Vanvitelli”, 80131 Naples, Italy; 6Oncology Laboratory, Foundation Casa Sollievo della Sofferenza IRCCS, San Giovanni Rotondo, 71013 Foggia, Italy; pparrella@operapadrepio.it; 7Division of Medical Genetics, Fondazione IRCCS-Casa Sollievo della Sofferenza, San Giovanni Rotondo, 71013 Foggia, Italy; a.petracca@operapadrepio.it

**Keywords:** *BRCA*, PARP inhibitors, HRD, immune checkpoint inhibitors, ICIs, ovarian cancer, OC, durvalumab, olaparib, niraparib

## Abstract

Ovarian cancer (OC) has a high impact on morbidity and mortality in the female population. Survival is modest after platinum progression. Therefore, the search for new therapeutic strategies is of utmost importance. *BRCA* mutations and HR-deficiency occur in around 50% of OC, leading to increased response and survival after Poly (ADP-ribose) polymerase inhibitors (PARPis) administration. PARPis represent a breakthrough for OC therapy, with three different agents approved. On the contrary, immune checkpoint inhibitors (ICIs), another breakthrough therapy for many solid tumors, led to modest results in OC, without clinical approvals and even withdrawal of clinical trials. Therefore, combinations aiming to overcome resistance mechanisms have become of great interest. Recently, PARPis have been evidenced to modulate tumor microenvironment at the molecular and cellular level, potentially enhancing ICIs responsiveness. This represents the rationale for the combined administration of PARPis and ICIs. Our review ought to summarize the preclinical and translational features that support the contemporary administration of these two drug classes, the clinical trials conducted so far, and future directions with ongoing studies.

## 1. Introduction

Ovarian cancer (OC) represents the eighth most common tumor among the female population, with an incidence of around 11 cases/100,000 women/year [1]. OC is the most lethal of gynecological tumors, with less than 25% of patients alive after five years from diagnosis at an advanced stage [1,2]. Almost all OC subtypes have an epithelial origin. Among them, high-grade serous OC (HGSOC) represents around 70% of cases, with aggressive features and often being metastatic at diagnosis [3]. Platinum-based chemotherapy represents a cornerstone of advanced OC treatment, still associated with a high relapse rate, particularly in the first two years. However, most patients will recur and develop resistant disease [4,5,6,7]. Thus, searching for new therapeutic strategies is mandatory for advanced OC. Several studies aimed to target specific pathways to improve efficacy results in advanced OC in the past decade. BReast CAncer gene (*BRCA*) mutations are detectable in almost 50% of OC patients, particularly in HGSOC [8,9]. Poly (ADP-ribose) polymerase (PARP) inhibitors (PARPis) have shown efficacy as maintenance in patients with platinum-sensitive recurrent OC (PS-ROC) and subsequently high grade OC patients that did not progress after first-line platinum-based chemotherapy [10,11,12,13,14,15,16,17,18].

In the same period, immune checkpoint inhibitors (ICIs) were reshaping the treatment scenario of many solid tumors, however this was not the case with OC. Tested mostly in pre-treated patients as single agents, ICIs showed modest response rates and survival results, and some phase III trials were even prematurely closed for futility [19,20,21,22]. Given these dismal results, combining immunotherapy and agents with different mechanisms of action appears to be a promising strategy for eliciting an immune response in advanced OC patients. One of the most intriguing combinations is with PARPis: some phase I/II trials have been conducted, and others are ongoing [23,24,25,26].

With our review, we sought to summarize the backstage for the combined use of PARPis and ICIs in OC, aiming to deepen the rationale underlying the combination of these two drug classes and its implications for clinical practice.

## 2. *BRCA* Mutations and PARPis in OC

### 2.1. The ‘Synthetic Lethality’ in OC

Double-strand DNA breaks (DSB) constitute the most severe type of DNA damage, as they disrupt both DNA reading frames, leading to mutations or chromosome rearrangements, increasing the oncogenic risk, and determining cell death [27]. Homologous recombination (HR) represents a key mechanism for DSB repair [27,28]. Other pathways, such as non-homologous end joining (NHEJ), are efficient in DNA repairing but also more error-prone, potentially causing DNA rearrangements [28]. HR is highly accurate for DSB repair, as an undamaged DNA template for neo-synthesis derives from a donor single-strand DNA (ssDNA) fragment [29]. *BRCA*
*1* and *2* are two essential proteins for the HR mechanism, as *BRCA1* is part of a surveillance complex for DSBs, and *BRCA2* cooperates with RAD51-Recombinase (RAD51) in repairing DSBs [30,31,32]. Several studies showed a survival advantage for *BRCA*-mutant patients with OC and a better response to DNA-damaging chemotherapeutic agents like platinum compounds [8,33]. Besides germline mutations of *BRCA1* and *2*, several genes confer a similar sensitivity to PARPis in case of somatic mutations, named under the term ‘BRCAness’. These include mutations in *BRCA1* and *BRCA2*, *RAD51* (locator of the DNA repair complex to the broken DNA strand), Ataxia-Telangiectasia Mutated (*ATM*), ATR Serine/Threonine Kinase (*ATR*—two DNA-damage sensory proteins), *BRCA1* Associated RING Domain 1 (*BARD1*), *BRCA1* Interacting Protein 1 (*BRIP1*), cyclin-dependent kinase 12 (*CDK12*), Partner and localizer of the *BRCA2* (*PALB2*), and Fanconi anemia complementation group (*FANC*—that constitutes a complex cooperating with *BRCA* in DNA repair) [34,35].

Single-strand DNA breaks (SSB) represent another type of DNA damage. SSBs are fixed by three mechanisms: base excision repair (BER), nucleotide excision repair (NER), and mismatch repair (MMR). The PARP family is a group of 17 proteins, of which PARP1-3 are deputed to repair DNA breaks through BER that supplies *BRCA* inefficiency. PARP 1 and 2 determine the poly-ADP ribosylation (called ‘PARylation’) of chromatin and auto-PARylation. PARP1 opens up chromatin and recruits factors to repair DNA [36,37]. PARP1 can recruit *BRCA1* for HR or NHEJ-associated factors. PARP2 is supposed to limit 53BP1 in favor of *BRCA1* accumulation, promoting HR over NHEJ [38]. Auto-PARylation is useful for releasing PARP from DNA binding and allowing DNA repairing proteins to access DNA and complete the repair process [36]. Therefore, since *BRCA* mutant cells are inefficient in HR, if PARP is blocked, SSBs are more likely to accumulate and generate potential DSBs, inducing a genomic instability ending in cell death: the so-called ‘synthetic lethality’ represents the rationale for PARPis use in *BRCA* mutant tumors, OC included [37]. Initial studies on synthetic lethality postulated that PARPis cause SSBs determining the collapse of the replication fork, inducing the more effective DSBs, the more defective the HR pathway is [39]. More recently, it has been demonstrated that PARP could be trapped in DNA by PARPis, which block both PARylation of downstream substrates and auto-PARylation, enhancing PARP1 avidity for DNA after allosteric changes in its structure [40]. As a result, the progression of the replication fork is stopped, resulting in a cytotoxic effect, as unrepaired SSBs convert into DSBs [39,41]. A further hypothesis is that PARPis enhance NHEJ, leading to further genomic instability and cell death [42]. Taken together, PARP trapping and NHEJ enhancement make the effect of PARPis stronger than PARP depletion [43]. All the clinical approved PARPis potently inhibit PARP1 and PARP2.

### 2.2. Clinical Applications of PARPis in OC

Currently, three different PARPis have been FDA/EMA approved in OC: olaparib, rucaparib, and niraparib. In OC, PARPis demonstrated efficacy in metastatic OC patients as maintenance after frontline chemotherapy or in the platinum-sensitive recurrent disease [11,12,13,14,15,16,17,18]. Starting from 2014, PARPis were introduced in the clinical practice for the treatment of PS-ROC, as the three randomized phase III trials NOVA, SOLO-2, and ARIEL3 demonstrated a significant progression-free survival (PFS) benefit (Hazard Ratio [HR] for PFS ranging from 0.12 to 0.54) with niraparib, olaparib, and rucaparib maintenance versus placebo (PBO), respectively. Median PFS (mPFS) ranged from 8.4 to 21 months with PARPis versus 3.8–5.5 months with PBO [10,11,12,13,14]. In the three phase III trials SOLO-1, PRIMA, and VELIA, the maintenance with PARPis demonstrated a reduction of the risk of progression between 32% and 80% compared to PBO in patients achieving a stability/response after frontline platinum-based chemotherapy [10,15,16,17]. In *BRCA* mutant patients, the benefit was particularly significant, as PFS was almost doubled with PARPis versus PBO (mPFS ranged from 22 to 37 months with PARPis versus 10–22 months with PBO). Moreover, in the PAOLA-1 trial, the combination of olaparib and bevacizumab determined a PFS advantage in HR-deficient (HRD) patients as maintenance after first-line platinum (37 vs. 17 months), enlightening the value of synthetic lethality when approaching these patients [18]. Of note, the above-mentioned studies were designed to assess PFS as the primary endpoint, and overall survival (OS) results are still ongoing [11,12,13,14,15,16,17,18].

However, more than 40% of OC patients failed to respond to PARPis, and different mechanisms have been addressed: the secondary reversion mutations in genes such as *BRCA1*, *BRCA2*, *RAD51*, *PALB2* that can restore the open reading frame; loss of p53 promoting NHEJ; mutations in PARP1 DNA-binding domain that increase auto-PARylation; modifications of regulators of the replication fork degradation such as Pax Transactivation-Domain Interacting Protein (PTIP) and Enhancer of zeste homolog 2 (EZH2) [44,45,46].

Many studies focused on overcoming resistance to single-agents PARPis for broadening the responding population. The immune response arising against dying tumor cells triggered by synthetic lethality, associated with neo-antigens release, led to the hypothesis that PARPis and ICIs could reciprocally potentiate, reducing resistance mechanisms. Further confirmations derived after a series of immunomodulant effects of PARPis were evidenced: the interaction with the tumor microenvironment (TME) of OC, the increased number of TILs, the upregulation of PD-L1, the enhanced antigen presentation and tumor mutational burden (TMB), and the interaction with the stimulator of interferon genes (STING) pathway [47,48,49,50,51,52].

## 3. Immune Checkpoint Inhibitors in OC

### 3.1. ICIs Pathways and Clinical Applications in OC

In the current clinical practice, immunotherapy mainly relies on ICIs, a group of antibodies disrupting the negative signals for the anti-tumor immune system, dampening cytotoxic T-cells [53]. Antigen-presenting cells (APCs) activate T-cells after presenting antigens bound to major histocompatibility complex (MHC) to T-cell receptor (TCR), together with co-stimulatory signals developing after cluster of differentiation (CD)-80/B7-1 and CD86/B7-2 on APCs bind CD28 located on T-cells. Subsequently, activated T-cells express co-inhibitory molecules, such as Cytotoxic T-lymphocyte-associated protein 4 (CTLA-4) and Programmed-Death (PD)-1 at immune checkpoints, whose equilibrium with the co-stimulatory signals is crucial for the correct activity and tolerance of T-cells [54]. CTLA-4 is constitutively expressed on Tregs and can be transported to the T-cell surface proportionally to antigen stimulation after antigen response. It binds B7 proteins with higher affinity than CD28, causing inactivation and anergy of T-cells [55]. Moreover, PD1 is widely expressed on immune cells, T-cells, B-cells, Natural killer (NK)-cells, and Dendritic cells (DCs). PD1 natural ligands are represented by Programmed Death Ligand (PD-L)1 and PD-L2. After TCR binds antigen presented by MHC, PD1 binds its ligands, becoming functional. Subsequently, a downstream inhibitory cascade is started, which stops the activation signal. Blocking CTLA-4 and PD1/PD-L1 axes, ICIs leverage on this balance, shifting the immune system towards the activation.

Almost 20 clinical trials explored the efficacy and safety of ICIs in the advanced setting of OC. The majority of studies were phase I/II trials. Patients were often heavily pre-treated. Anti-PD1 (pembrolizumab, nivolumab), anti-PD-L1 (avelumab, atezolizumab, durvalumab), and anti-CTLA4 (ipilimumab, tremelimumab) agents were administered as monotherapy or in combination. Response rates and survival were unsatisfactory (overall response rate [ORR] < 10% with single-agents ICIs, reaching 47% with the addition of chemotherapy and bevacizumab; no survival differences), leading to the premature stopping of some phase III trials (reviewed in [19]). Only three phase III trials have been completed [20,21,22]. In JAVELIN 200 (NCT02580058), avelumab did not improve OS or PFS versus chemotherapy when used as a single agent and combined with chemotherapy [20]. In the IMagyn050, adding avelumab to chemotherapy and bevacizumab did not improve survival in the first-line (*p* = 0.28) [21]. In the NINJA trial (conducted in the Japanese population), no OS differences emerged between nivolumab and chemotherapy in patients with recurrent OC (HR = 1.03) [22].

### 3.2. Predictive Factors for ICIs in OC

A possible reason for these unsatisfactory results relies on different immunosuppressive factors within the OC TME. A series of cytokines, such as interferon-gamma (IFNγ), interleukin (IL)-6, IL-10, transforming growth factor-beta (TGFβ), and tumor necrosis factor-alpha (TNFα), induce immunosuppressive cells, such as myeloid-derived suppressor cells (MDSCs), and polarize tumor-associated macrophage (TAMs) towards the immunosuppressive M2 subtype. Effectively, M2 macrophages emerged as the predominant TAM subpopulation in OC, associated with an advanced stage and a negative prognostic role [56,57]. MDSCs inhibit T-effectors and NK-cells. Moreover, these cytokines induce cyclooxygenase-2 (COX-2) production, leading to Prostaglandin E2 (PGE2) synthesis and resulting in limited recruitment of T-cells at tumor sites. On their way, M2 macrophages produce cytokines that inhibit T-effectors and enhance Tregs (IL-1R, IL-10, C-C-Motif Chemokine Ligand [CCL] 17, CCL20, CCL22) [58,59]. Tregs produce IL10 and TGFβ that, together with other immunosuppressive cytokines such as IL6, inhibit T effectors and reduce DCs and APCs activity [60]. DCs can recognize damage-associated molecular patterns (DAMP) released from dead OC cells and activate CD4^+^ and CD8^+^ via MHC class I and II, respectively [61]. In OC patients, a tolerogenic DC group has been found, associated with lower levels of pro-inflammatory cytokines, but the release of enzymes that reduce T-effectors activity such as Indoleamine 2,3-Dioxygenase (IDO). Moreover, in mouse models of OC, it has been evidenced that, when the tumor stage increases, DCs gradually assume an immunosuppressive phenotype [62].

After years of ICIs use, it is well-known that no unique predictive biomarker exists. However, several factors can influence the response to ICIs: neo-antigens production and tumor mutational burden (TMB), number of TILs, TME, mismatch-repair deficiency (MMRd), leading to microsatellite instability (MSI) [63]. This led to the development of ICIs-based combination therapies, among which PARPis represent an option.

## 4. Rationale to Combine PARPis and ICIs in OC

### 4.1. PD-L1 Upregulation

Historically, PD-L1 has been indicated as the first biomarker for ICIs response, even if its cut-off and detection method remain blurred across different studies and tumor subtypes also in OC [19].

PARPis have been associated with an increased PD-L1 expression [47]. It became clear in murine models after the administration of PARPis, such as olaparib and talazoparib, which increased PD-L1 levels. This mechanism seems to be driven by the inactivation of glycogen synthase kinase 3-beta (GSK3β) by PARPis: GSK3β induces proteasomal degradation of PD-L1, thus modulating its surface expression [64]. As proof, in cell lines and murine models knockout for GSK3β, PD-L1 did not increase after treatment with olaparib [47]. Another way used by PARPis to upregulate PD-L1 is through the STING pathway, in response to IFNγ secretion [65]. Finally, the third way PARPis upregulate PD-L1 is ATM-ATR-Checkpoint kinase 1 (CHEK1) [66]. ATM acts as a kinase sensor for DSBs [67]. After ATM is activated, a switch in a signal kinase from ATM to ATR occurs, during which ATM activity is progressively attenuated in favor of ATR. Finally, the ATM-to-ATR switch activates Chk1 [68]. As a result, the JAK/STAT signaling, which increases PD-L1 expression, is activated [66,68]. The role of the ATM-ATR-CHEK1 pathway in upregulating PD-L1 expression on the cell surface was confirmed after the administration of ATM- or CHK1-inhibitors in cell models. It is noteworthy that the combination of ICIs and PARPis leads to a greater activation than the single agent [66]. Therefore, we can assume that, increasing PD-L1 on the cell surface, PARPis contribute to a higher ICIs response (Figure 1).

### 4.2. Interactions between PARPis and TME

TME plays a crucial role in the response to therapies. PARP and *BRCA* pathways interact with the TME, as they are involved in the correct development of T-cells, inflammatory and immune responses development, and expression and regulation of both soluble factors and cellular components within the TME [69,70,71,72]. Regarding T-cells, *PARP2* seems involved in T-cell maturation, *PARP1* regulates the differentiation of T-effectors versus Tregs through the expression of FoxP3, CD4^+^ differentiation, and TGFβ expression, both *PARP1* and *2* regulate T-cells function [73]. Regarding APCs, *PARP1* has been involved in DCs recruitment and functioning, perhaps through the activation of the STING pathway in DCs [74,75]. These genes influence other cells in TME: *PARP1* seems associated with macrophages polarization, and increased MDSCs have been found in *BRCA1* mutant models [72,76]. Indeed, immune-suppressive elements such as MDSCs or TAMs are recruited at sites where continuous low-level DNA damage sustains a sort of chronic inflammation. In turn, MDSCs and TAMs promote further DNA damage through free radicals generation [77].

The role of PARPis is essentially to shift this sort of chronic inflammation to a more immune-responsive TME through widespread effects on cells involved in innate and adaptive immune response and soluble factors [48]. PARPis block the transcription of lipopolysaccharide-induced macrophage cytokines and other pro-inflammatory cytokines promoting TAMs differentiation towards the M2 subtype, associated with tumor invasion and spread [78,79]. PARPis enhance APCs by upregulating MHC after activating ATM/ATR kinases, leading to CD8^+^ and CD4^+^ T-cells activation [51,74,80]. On the contrary, inhibitory factors such as PD1, T cell immunoglobulin, mucin domain-containing protein 3 (TIM-3), and Lymphocyte-activation gene 3 (LAG-3), or MDSCs, were reduced in mouse models or murine orthotopic cancer cell lines when PARPis and anti-PD1 were co-administered, leading to a higher ICIs responsivity [48,69,72,81] (Figure 2).

### 4.3. TILs Increase

In OC patients, *BRCA* mutations have been associated with a more robust infiltrate of TILs, having a good prognostic value [49,82]. *BRCA* mutations induce higher TILs infiltration than other HR genes defects, and even more than HR-proficient tumors expressing low TILs and immune checkpoint regulators such as PD1 and PD-L1 [49]. High TILs, especially CD8^+^ T-cells, are associated with better prognosis in OC, whereas high Tregs have a negative prognostic meaning [83,84]. TILs infiltration has been indicated as a possible biomarker for ICIs sensitivity. As evidenced in mouse models, the combination of PARPis and ICIs increases TILs and PD-L1 expression via GSK3β, overcoming ICIs resistance [47]. Following an IFN-I response, chemokines are responsible for T-cell recruitment at tumor sites, increasing TILs. Acting through the Nuclear factor of activated T cells (NFAT), PARPis promote Th1 cells, enhancing APCs [73] (Figure 1 and Figure 2). Moreover, interacting with Nuclear factor-kappa B (NF-kB), epithelial-to-mesenchymal transition (EMT) is inhibited by PARPis, therefore increasing immune cells’ penetration at tumor sites [85].

### 4.4. Neo-Antigens and TMB Increase

In many solid tumors, the mutation in genes involved in DNA repair is associated with a good ICIs response, as it enhances TMB and neo-antigens production [75]. For its part, TMB further increases neo-antigens production and TILs [66]. Even if a cut-off for TMB has not been established, it is considered a surrogate of neoantigen load, which is associated with a good response to ICIs [86,87].

In murine models, *BRCA* mutations have been associated with a high TMB, inducing tumor cell damage, increasing neo-antigens production, and T-cell responses that are amplified by ICIs [50,75]. From the analysis of human OC samples of The Cancer Genome Atlas (TCGA) database, a similar neo-antigens load is detected in the case of *BRCA1* and *BRCA2* mutations. The neo-antigens burden is lower in HRD, but still higher than HR-proficient tumors [3]. Additionally, the mutations in other genes involved in DNA repair, such as *MSH2* and *POLE*, correlate with neo-antigens production and ICIs response. Effectively, a higher TMB corresponds to a higher expression of genes involved in immune response, such as the TCR signaling, IFNγ, and TNF receptors. Moreover, OC carrying a higher TMB expresses higher PD-L1 levels [82]. PARPis induce DNA damage, leading to neo-antigens production, justifying the potential effectiveness of PARPis and ICIs combinations to reciprocally potentiate the two mechanisms and broaden the responders’ audience. Indeed, after PARPis administration, DNA damage results in the accumulation of DNA fragments within the cytoplasm. Thus, a higher number of neo-antigens is produced and exposed to the cell surface, increasing immune response activation, and resulting in higher TMB, with higher immunogenicity and therefore better possibility for ICIs response in OC [49,75,88] (Figure 1 and Figure 2).

### 4.5. STING Pathway

Another mechanism of interaction between PARPis and the immune system is through the pathway of STING, a system involved in the production of IFNγ and pro-inflammatory cytokines [74,89] (Figure 1). After PARPis induce DNA damage, DNA fragments accumulate in the cytoplasm and are recognized by cytosolic sensor cyclic GMP-AMP synthetase (cGAS). cGAS activates the downstream 2′-5′ cyclic GMP-AMP (cGAMP), a second messenger that switches on STING. STING stimulates the transcription factors NF-kB, and IFN regulatory factor 3 (IRF3). After phosphorylation and Nuclear translocation, NF-kB and IRF3 regulate the transcription of type I IFN [61,90,91]. IFN-upregulation promotes an immune response through the chemotaxis of T-cells, NK cells, and DCs [52,92]. In an autocrine or paracrine manner, IFN stimulates the Janus kinase/signal transducer and activator of transcription proteins (JAK/STAT) pathway, leading to the expression of IFN-related genes [93]. Additionally, as aforementioned, STING upregulates PD-L1 expression [66,67]. As proof, in *BRCA*-mutant cancer cells engrafted in mice models, the activation of STING was linked with a series of immunological changes after olaparib administration: CD3^+^, CD4^+^, and CD8^+^ T-cells, NK cells, and circulating levels of pro-inflammatory cytokines such as IFN, CCL5, and C-X-C motif chemokine ligand 10 (CXCL10) increased [74]. In case of STING depletion, the immunostimulant activity of olaparib did not occur [51]. These effects were further enhanced by the co-administration of ICIs, once again reinforcing the rationale for PARPis and ICIs combination [52].

## 5. Clinical Trials of PARP Inhibitors and ICIs Combination

As PARPis revolutionized the therapeutic scenario of OC, whereas ICIs did not result in efficacy improvement, more recently, overcoming resistance to ICIs as a single agent with PARPis combination has appeared as an attractive strategy. So far, four clinical trials investigating the efficacy of the association of ICIs and PARPis have been published (Table 1).

In the phase I-II TOPACIO/Keynote-162 trial (NCT02657889), 62 women with recurrent platinum-resistant/refractory OC were selected to receive the PARPi niraparib (200 mg once daily) plus the anti-PD1 pembrolizumab (200 mg every 3 weeks [q3w]). Patients were not selected for *BRCA* or *HR* status: the majority of them were *BRCA* wild type (wt-79%) or *HR* proficient (53%). ORR and disease control rate (DCR) were the primary endpoints. Treated patients reached an ORR of 25%, and a DCR of 68%. Eight patients achieved a complete response (CR), of which 5 were *BRCA*wt. Among 11 *BRCA*-mutant patients, an ORR of 45%, and a DCR of 73% were reached [23]. The combination of *HR* proficiency and an interferon-primed exhausted CD8^+^ T-cells score positivity was associated with objective response [94].

In the phase II NCT02484404 study, 35 patients with pretreated ROC, predominantly PR-ROC (30/35—86%) and *BRCA*wt (27/35—77%), were recruited to receive olaparib (300 mg twice daily) plus the anti-PD-L1 durvalumab (1500 mg q4w). ORR—the primary endpoint—was 14% (95% CI, 4.8–30.3%) with 5 partial responses (PR), thus not meeting the pre-specified primary endpoint of at least 17%; DCR was 71% (95% CI, 53.7–85.4%); mPFS was 3.9 months. It is also worthy of note that this treatment enhanced an immunostimulatory environment, increasing IFNγ, TNFα, CXCL9/CXCL10, and TILs. Moreover, PFS was significantly correlated with an increased IFNγ production (*p* = 0.023), whereas a worse PFS was associated with high Vascular Endothelial Growth Factor Receptor (VEGFR)-3 levels [24].

In the phase II MEDIOLA (NCT02734004) trial, a total of 95 patients with PS-ROC were recruited: 32 of them had germline *BRCA* mutations (g*BRCA*m), 63 were *BRCA*wt. 32 g*BRCA*m patients received olaparib for 4 weeks followed by olaparib plus durvalumab q4w. ORR was 71.9%, mPFS was 11.1 mos, and 12-wks DCR was 81%. Seven out of thirty-two patients (21.8%) had a CR. The second group of 32 *BRCA*wt patients received durvalumab plus olaparib, reaching an ORR of 31.3%, an mPFS of 5.5 mos, and a 24-wks DCR of 28.1%. Finally, 31 women with *BRCA*wt received olaparib plus durvalumab plus bevacizumab: ORR of 77.4%, mPFS of 14.7 mos, and 24-wks DCR of 77.4% were observed [25].

The combination of PARP-inhibition and CTLA-4 blockade has been investigated in the NCT02571725 phase Ib/II study, with only 3 g*BRCA*mut recurrent OC patients treated with olaparib (300 mg twice daily) and tremelimumab (10 mg/kg q4w for two cycles). Three PRs were observed, with significant reductions of Ca125 and a decrease in tumor size [26].

Despite OS data being immature, the trials conducted so far did not evidence a significant difference in response rates and survival according to *BRCA*/HR status, differently from *BRCA* single agents that were significantly more effective in *BRCAm*/*HRD* patients [10,11,12,13,14,15,16,17,18]. Therefore, the role of *BRCA* mutations seems to be downgraded when considering the association of PARPis and ICIs. Ongoing trials will further investigate this hypothesis (Table 2).

Defects of genes involved in mismatch repair (MMR), such as *MSH2*, *MSH6*, and *MLH1*, leading to microsatellite instability (MSI), have been found in 2–20% of OC [95]. Tumors carrying MSI express features of high immune responsiveness: in fact, pembrolizumab was FDA-approved as the first agnostic therapy for these subsets of tumors [96]. In immunoprecipitation and mass spectrometry analysis, it has been identified that *BRCA1* is often associated with other proteins involved in DNA damage repair in a complex named BASC (standing for ‘*BRCA1* associated genome surveillance complex’), which acts as a sensor for DNA damage [97]. Proteins involved in MMR are part of this complex, therefore we can hypothesize that DNA damage, sustained by PARPis, is further reinforced by MMR. Another topic of interest is the real advantage of combining PARPis plus ICIs versus ICIs in such an immune-prone setting.

In light of this, future trials will point out intriguing questions, such as the optimal setting for the combination, between first-line, PS-ROC, platinum-resistant ROC, and even an earlier approach in the preoperative stage, and the molecular characteristics which will confer an advantage to the combination over single agents. Moreover, phase III trials will compare different treatment arms, overcoming the absence of randomization that represents a limitation of the studies published so far. These studies will also try to shed light on the possibility of combining PARPis and ICIs with different drugs. For example, the PARP pathway interacts with Hypoxia-inducible factors (HIF)-1α and -2α, driving an angiogenic response to hypoxia [98]. Indeed, the combination of olaparib and bevacizumab resulted in effective first-line maintenance [17]. Similarly, there is an intensely studied relationship between the immune system and angiogenesis, as ICIs and anti-angiogenic drugs are already approved for the treatment of other tumor subtypes, gynecological cancers included [19,99]. Angiogenesis is indeed a key mechanism for OC proliferation, with the demonstrated efficacy of the anti-VEGF bevacizumab [100]. ICIs alone did not improve efficacy if added to anti-angiogenics, as demonstrated by the IMagyn050 trial in newly diagnosed OC [20]. However, the addition of PARPis to ICIs and anti-angiogenics would be intriguing (Table 2). Over efficacy, concerns in these combinations remain regarding safety. Although ICIs did not generally worsen toxicity, the dosage and duration of combination treatment were heterogeneous in the trials conducted so far, and future trials will further investigate the optimal schedule [23,24,25,26].

## 6. Conclusions

Genomic instability resulting from *BRCA* and HR mutations represents one of the hallmarks of cancer [101]. On their way, ICIs have revolutionized the anticancer therapeutic algorithm of the last ten years in many solid tumors. When used separately, PARPis are of utmost efficacy in OC, with multiple drugs approved in the clinical practice; on the contrary, ICIs trials have not modified the OC therapeutic sequencing because of poor response rates and survival outcomes. The immunomodulant effect of PARPis represents a rationale for combination with ICIs. Effectively, this combination appears synergistic and represents a compelling strategy for advanced OC therapy. Significant pre-clinical evidence supports this strategy. However, before considering the association as a critical change in the management of OC patients, several questions remain unanswered. A critical step will be to identify the optimal patients that will most likely benefit from the association, the role that *BRCA* mutations or HRD will play in front of these combinations, and potential differences between somatic and germline mutations. Biomarkers for the combination should be investigated. For this purpose, thriving research in other solid tumors is trying to identify predictive factors, which similarly could be of interest in OC. The appropriate dosage of each agent and duration of the regimens should be determined to maximize both efficacy and safety. Finally, clarifying the resistance mechanisms to each agent will further help the correct sequencing strategy of the combination.

## Figures and Tables

**Figure 1 ijms-23-03871-f001:**
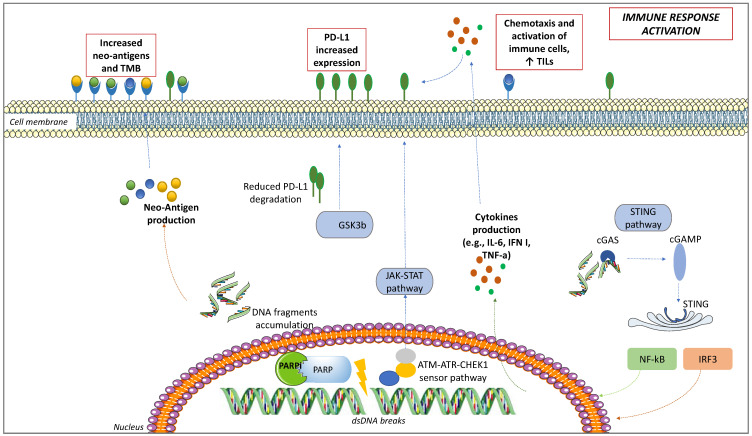
The interplay between PARP inhibitors (PARPis) and immune checkpoint inhibitors (ICIs). When double-strand DNA breaks (DSBs) occur and PARPis block PARP complex, DNA fragments accumulate in the cytoplasm. As a result, neo-antigens accumulate on the cell surface, where they are recognized by antigen-presenting cells (APCs), activating the immune response. Moreover, the Stimulator of interferon genes (STING) pathway is activated: DNA fragments are recognized by cytosolic sensor cGMP-AMP synthetase (cGAS), cGAS activates 2′-5′ cyclic GMP-AMP (cGAMP), cGAMP switches on STING, STING modulates transcription factors such as Nuclear factor-kappa B (NF-kB), and Interferon regulatory factor 3 (IRF3), resulting in the transcription of related cytokines (Interferon [IFN], IL-6, tumor necrosis factor-alpha [TNFα]), promoting immune response. IFN increases the expression of Programmed Death-Ligand 1 (PD-L1) on the cell surface. The activation of the sensor system Ataxia-Telangiectasia Mutated (ATM)-ATR Serine/Threonine Kinase (ATR)-Checkpoint kinase 1 (CHEK1) in case of DSBs, and the STAT-IRF pathway, increase PD-L1. Finally, DSBs inactivate glycogen synthase kinase 3-beta (GSK3β), responsible for PD-L1 proteasomal degradation, increasing PD-L1 cellular expression. These modifications result in a more immune responsive TME: increased surface neo-antigens, increased PD-L1 expression, cytokines, and chemotactic factors determine an increase in number and function of APCs, T-cells, NK cells, and decrease in immunosuppressive elements such as myeloid-derived suppressor cells and M2 macrophages.

**Figure 2 ijms-23-03871-f002:**
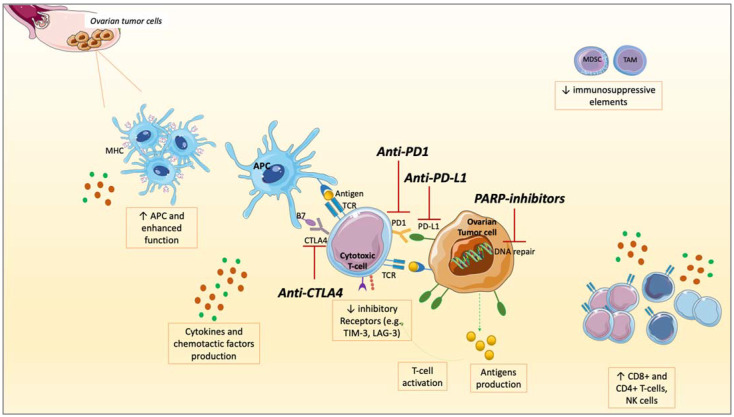
The interplay between PARP inhibitors (PARPis) and immune checkpoint inhibitors (ICIs): modifications of soluble factors and cell of tumor microenvironment (TME) in ovarian cancer (OC). After the administration of PARPis, a series of modifications occur: an increased production and surface exposure of antigens on tumor cells, increased PD-L1 expression, production of cytokines, and chemotactic factors. Antigen-presenting cells (APCs) are enhanced in their function with major histocompatibility complex (MCH) up-regulation and increased in numbers. A higher number of CD4+ and CD8+ T-cells, and NK cells, are recruited at tumor sites, resulting in higher TILs and immune response activation. On the other hand, immune-suppressive elements such as Myeloid-derived suppressor cells (MDCSs) and Tumor-associated macrophages (TAMs) are reduced, as well as inhibitory receptors (such as T-cell membrane protein 3 [TIM-3], Lymphocyte-activation gene 3 [LAG-3]). These modifications shift the TME toward a higher immune responsivity. Targeting Programmed Death-Ligand 1 (PD-L1), PD1, or Cytotoxic T-lymphocytes Associated Protein 4 (CTLA4), ICIs unleash the anti-tumor immune response, potentiating the immune activation against tumor cells. Therefore, combining these two drug classes could result in a higher anti-tumor immune response in OC.

**Table 1 ijms-23-03871-t001:** Clinical trials of PARPis plus ICIs combination in OC with published results.

Study Name/NCT Identifier	Phase	Target Population (*n*)	ICI (Target)	PARPi	Results
TOPACIO/Keynote-162/NCT02657889 [23]	I-II	PR-ROC (*n* = 62), not selected for *BRCA*/HR status	Pembrolizumab 200 mg q3w (anti-PD1)	Niraparib 200 mg OD	ORR 25%, DCR 68% *BRCA*m (21%): ORR 45%, DCR 73% 8 CR (5 *BRCA*wt patients)
NCT02484404 [24]	II	ROC (*n* = 35: 30 PR-ROC + 5 PS-ROC): *BRCA*wt (*n* = 27) g*BRCA*mut (*n* = 6) s*BRCA*mut (*n* = 2)	Durvalumab 1500 mg q4w (anti-PD-L1)	Olaparib 300 mg BID	ORR 14%, DCR 71% 5 PRs (2 g*BRCA*mut, 2 *BRCA*wt, 1 s*BRCA*mut) mPFS 3.9 mos
MEDIOLA/NCT02734004 [25]	II	g*BRCA*mut PS-ROC (*n* = 32)	Durvalumab 1500 mg q4w	Olaparib 300 mg BID	ORR 71.9% mPFS 11.1 mos 12-wks DCR 81% 7/32 CR (21.8%)
		PS-ROC *BRCA*wt (*n* = 32)	Durvalumab	Olaparib	ORR 31.3%, 24-wks DCR of 28.1% mPFS 5.5 mos
		PS-ROC *BRCA*wt (*n* = 31)	Durvalumab	Olaparib + Bevacizumab 10 mg/kg q2w	ORR 77.4%, 24-wks DCR 77.4% mPFS 14.7 mos
NCT02571725 [26]	Ib/II	g*BRCA*mut ROC (*n* = 3)	Tremelimumab 10 mg/kg q4w (anti-CTLA4)	Olaparib 300 mg BID	ORR 100% 3 PRs

BID: bis-in-die; *BRCA*: Breast Cancer gene; *BRCA*wt: *BRCA* wild type; CTLA4: cytotoxic T-lymphocyte-associated protein 4; CR: complete response; DCR: disease control rate; g*BRCA*mut: germline *BRCA* mutant; HR: homologous recombination; HRD: HR defective; OD: once daily; ORR: overall response rate; PD1: programmed death-1; PD-L1: programmed death-ligand 1; PFS: progression-free survival; PR: partial response; PR-ROC: platinum resistant recurrent ovarian cancer; PS-ROC: platinum sensitive recurrent ovarian cancer; q3(4)w: every 3(4) weeks; ROC: recurrent ovarian cancer; s*BRCA*mut: somatic *BRCA* mutant.

**Table 2 ijms-23-03871-t002:** Ongoing trials of PARPis and ICIs combination in OC.

Clinicaltrials. Gov Registration/NAME	Phase	Setting	Combination
NCT03602859/FIRST	III	First-line, Stage III/IV EOC	Niraparib + Dostarlimab
NCT03522246/ATHENA	III	Maintenance after first-line platinum-based CT	Rucaparib + Nivolumab
NCT04679064/NItCHE-MITO33	III	Platinum-ineligible ROC	Niraparib + Dostarlimab
NCT04361370/DUO-O	III	Maintenance after first-line platinum-based CT	Olaparib + Durvalumab + Bevacizumab
NCT03740165/KEYLYNK-001	III	Maintenance after first-line platinum-based CT	Olaparib + Pembrolizumab
NCT02873962	II	ROC	Rucaparib + Nivolumab + Bevacizumab
NCT03955471/MOONSTONE	II	PS-ROC	Niraparib + Dostarlimab
NCT03695380 (Cohort 2)	I/II	PS-ROC	Niraparib + Cobimetinib + Atezolizumab
NCT04673448	I	*BRCA*mut-ROC	Niraparib + Dostarlimab
NCT03651206/ROCSAN	II	Ovarian Carcinosarcoma	Niraparib + Dostarlimab
NCT04361370/OPEB-01	II	Maintenance in PS-ROC *BRCA*wt	Olaparib + Pembrolizumab + Bevacizumab
NCT04417192/OLAPem	II	Neoadjuvant OC	Olaparib + Pembrolizumab
NCT02953457	II	*BRCA*mut-ROC	Olaparib + Durvalumab + Tremelimumab

*BRCA*mut: breast cancer mutant; EOC: epithelial ovarian cancer; PS-ROC: platinum sensitive recurrent ovarian cancer; ROC: recurrent ovarian cancer.

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
