# Peer review of "The Interplay between PARP Inhibitors and Immunotherapy in Ovarian Cancer: The Rationale behind a New Combination Therapy"

_ijms, 2022, doi:10.3390/ijms23073871_

Round 1

Reviewer 1 Report

The manuscript entitled: "The interplay between PARP inhibitors and immunotherapy in ovarian cancer" emphasize the activity of PARP inhibitors on antitumor immune response aiming to bring towards clinical applications the combined use of PARP inhibitors together with checkpoint inhibitors. The molecular mechanisms beneath PARP inhibitor therapy were explained, as well as the premises of immune therapy in ovarian cancer; the authors made a bridge between the two concepts.  The results of the study are important not only from the viewpoint of the fundamental antitumor research, but could help to explain the results of ongoing trials and eventually to accelerate the immune therapy introduction to ovarian cancer treatment.

I recommend some minor revisions in the manuscript:

Abstract- the term PARP should be explained in the abstract too.

Introduction- reference is needed for:  "One of the most intriguing combinations is with PARPis: some phase I/II 59 trials have been conducted, and others are ongoing"

Chapter 2, lines 80-88- all of the itemized markers are relevant in ovarian cancer?

Line 129: "(..) series of immunomodulant effects of PARPis were evidenced: the 129 interaction with the tumor microenvironment (TME) of OC, the increased number of TILs, 130 the upregulation of PD-L1, the enhanced antigen presentation and tumor mutational bur-131 den (TMB), the interaction with the Stimulator of interferon genes (STING) pathway"- the appropriate references needs to be inserted

Chapter 3, line 137: please reformulate more clearly, for readers who are not familiar with the insides of costimulation via CD28 receptor.

From line 153: "Almost 20 clinical trials..." Although excellently synthesized by the authors in their previous review paper, the relevant trials have to be inserted here, together with the references, a brief description. Which PARP inhibitors were used in these trials?  

Chapter 4

Figure 1 - the letters and characters are very small inside of the image.

Line 294, please reformulate: "It activates the downstream 2’-5’ cyclic GMP-AMP 294 (cGAMP), which switches on STING. STING stimulates NF-kB, TANK-binding kinase 1 295 (TBK1), and IFN regulatory factor 3 (IRF3)."

Chapter 5- Table 1 correspond to completed trials?  Please point out.

Conclusions: "Effectively, the combination of PARPis and ICIs appears synergistic and represents a compelling strategy for advanced OC therapy" ? authors should mention here that the pre-clinical studies (chapter 4) support this statement.  

"Biomarkers for the combination should be investigated"- taking into account what authors stated in chapter 3 (179-183), the attempts to identify such markers in other solid tumors (lung, colon, melanoma, and others) could be extended to ICIs of OC?

Author Response

We would like to thank the reviewer for the valuable comments.

We modified the manuscript following the suggestions:

We introduced the explanation for PARP in the Abstract. 

In the Introduction, we added the reference as indicated.

In Chapter 2, we removed the non-relevant items for ovarian cancer, and added the references where indicated.

In Chapter 3, we re-formulated the period and added the relevant references of ICIs; the combined PARPis are mentioned in the dedicated chapter of the manuscript.

We modified the letters in Figure 1 to a larger size.

We reformulated the indicated period in Chapter 4.

We indicated that the trials in Table 1 are published.

We added the suggested statements in the Conclusions.

Reviewer 2 Report

This manuscript summarized the principles, clinical applications and future prospects of PARP inhibitors and immunotherapy with their combination.

The manuscript is well organized with a clear description. There are some concerns.

  1. It would be good that the title reflects the main hypothesis.
  2. There should be added references. 

             Incidence of ovary cancers (line 38)

             PARPis combination trials (line 60)

     3. Authors need to describe the detailed results on the treatment of PARPis, resistance rate of PARPis and  ICI results on ovary cancer.

     4. It would be good that the size of letter in Figure 1. is large.

     5. It would be good that figures about the mechanisms of PARPis & ICI add.

Author Response

We would like to thank the reviewer for the valuable comments.

We modified the title to reflect the aim of the review.

We added the missing references.

We added details regarding studies of PARPis and ICIs in the dedicated sections.

We modified the characters in Figure 1 to a larger size.

We added a Figure regarding the mechanisms of action of PARPis and ICIs.